# Analytical Method Development and Dermal Absorption of 4-Amino-3-Nitrophenol (4A3NP), a Hair Dye Ingredient under the Oxidative or Non-Oxidative Condition

**DOI:** 10.3390/toxics12050340

**Published:** 2024-05-07

**Authors:** Hyang Yeon Kim, Yu Jin Kim, Jung Dae Lee, Hak Rim Kim, Dong-Wan Seo

**Affiliations:** 1College of Pharmacy and Center for Human Risk Assessment, Dankook University, 119 Dandae-ro, Cheonan 31116, Chungnam, Republic of Korea; ahtsksdl32@naver.com (Y.J.K.); ljd0734@nate.com (J.D.L.); dwseomb@dankook.ac.kr (D.-W.S.); 2Center for Human Risk Assessment, Dankook University, Cheonan 31116, Chungnam, Republic of Korea; hrkim@dankook.ac.kr; 3College of Medicine, Dankook University, 119 Dandae-ro, Cheonan 31116, Chungnam, Republic of Korea

**Keywords:** 4-amino-3-nitrophenol (4A3NP), LC-MS, MRM, in vitro dermal absorption

## Abstract

The chemical 4-amino-3-nitrophenol (4A3NP) is classified as an amino nitrophenol and is primarily utilized as an ingredient in hair dye colorants. In Korea and Europe, it is exclusively used in non-oxidative or oxidative hair dye formulations, with maximum allowable concentrations of 1% and 1.5%, respectively. Despite this widespread use, risk assessment of 4A3NP has not been completed due to the lack of proper dermal absorption data. Therefore, in this study, both the analytical method validation and in vitro dermal absorption study of 4A3NP were conducted following the guidelines provided by the Korea Ministry of Food and Drug Safety (MFDS). Before proceeding with the dermal absorption study, analytical methods were developed for the quantitation of 4A3NP through multiple reaction monitoring (MRM) via liquid chromatography-mass spectrometry (LC-MS) in various matrices, including swab wash (WASH), stratum corneum (SC), skin (SKIN, comprising the dermis and epidermis), and receptor fluid (RF). These developed methods demonstrated excellent linearity (R^2^ = 0.9962–0.9993), accuracy (93.5–111.73%), and precision (1.7–14.46%) in accordance with the validation guidelines.The dermal absorption of 4A3NP was determined using Franz diffusion cells with mini-pig skin as the barrier. Under both non-oxidative and oxidative (6% hydrogen peroxide (H_2_O_2_): water, 1:1) hair dye conditions, 1% and 1.5% concentrations of 4A3NP were applied to the skin at a rate of 10 μL/cm^2^, respectively. The total dermal absorption rates of 4A3NP under non-oxidative (1%) and oxidative (1.5%) conditions were determined to be 5.62 ± 2.19% (5.62 ± 2.19 μg/cm^2^) and 2.83 ± 1.48% (4.24 ± 2.21 μg/cm^2^), respectively.

## 1. Introduction

The chemical 4-amino-3-nitrophenol (4A3NP) is classified as an amino nitrophenol and is primarily utilized as an ingredient in hair dye formulations, imparting colors ranging from bordeaux to red hues. It has been registered under the Registration, Evaluation, Authorization, and Restriction of Chemicals (REACH) regulation and imported at a volume from ≥1 to <10 tons per year in Europe. In Korea and Europe, it is solely employed in non-oxidative or oxidative hair dye formulations, with maximum allowable concentrations of 1% and 1.5%, respectively.

Overall, 4A3NP exhibits a variety of toxic effects in skin sensitization and gene mutation. In a sensitization study conducted on CBA/J mice, the stimulation index (SI) showed a dose-dependent increase ranging from 0.05% to 2.5%, with an EC3 value (the concentration inducing a three-fold increase in lymph node cell proliferative activity) of 0.2%, indicating extreme sensitization potential. Conversely, in a guinea pig maximization test (GPMT) involving female Hartley albino guinea pigs, no cutaneous reactions such as erythema or edema were observed following occlusive patch application, leading to the conclusion that 4A3NP was not sensitizing to guinea pig skin. Furthermore, in a gene mutation study, 4A3NP induced gene mutations in *Salmonella typhimurium* strain TA98, with metabolic activation and chromosome aberrations in cultured human lymphocytes with or without metabolic activation, whereas, micronucleated polychromatic erythrocytes did not show a significant increase with exposure to 4A3NP [1,2]. In the risk assessment report, the Scientific Committee on Consumer Products (SCCP, predecessor of the SCCS) suggested that 4A3NP is an extreme sensitizer. However, regarding genotoxicity, no conclusion could be reached, and additional experiments were recommended.

Since 2003, approximately 100 hair dye colorants have been assessed for their toxicological risk by the Scientific Committee for Consumer Safety (SCCS) of the EU Commission and have been permitted for use as cosmetic ingredients under Annex III of the EC Cosmetics Regulation. Risk assessment involves systematically identifying the toxicity of chemicals and evaluating the risks associated with their use. This process typically includes hazard identification, hazard characterization, exposure assessment, and risk characterization [3]. In the risk assessment of cosmetic ingredients, dermal absorption is crucial for calculating the systemic exposure dosage (SED) during exposure assessment [3]. However, in vitro percutaneous absorption studies of 4A3NP using human dermatomed skin were deemed inadequate due to the low number of chambers and high variability of skin penetration [2,4]. Therefore, a dermal absorption study is necessary for the risk assessment of 4A3NP.

In this study, a Franz diffusion cell system with a static fluid type was utilized under hair dye usage conditions to determine the dermal absorption rate of 4A3NP [5,6]. This system is well established for in vitro dermal absorption studies, considering the process of dermal absorption through the skin into the body using either real or recombinant skin [7,8]. The Franz diffusion cell consists of two compartments: the donor compartment, where the formulation dose is applied onto the skin, and the receiver compartment, which is filled with a solution circulated at a constant temperature to collect samples at different exposure times. Within the Franz diffusion cell system, there are two main methods: flow-through and static fluid cell systems [9]. While the flow-through system mimics the environment of blood circulation in the human body, the static fluid type offers advantages such as simplicity and a lower cost.

In previous skin absorption studies on 4A3NP, radioactive isotopes were utilized, and each compartment was measured using liquid scintillation counting [2,4]. In this study, 4A3NP in each matrix was quantified using multiple reaction monitoring (MRM) via liquid chromatography-mass spectrometry (LC-MS), a highly sensitive and selective analytical technique used for quantifying the target compound within complex samples [10,11,12]. In several papers or reports, 4A3NP was analyzed alongside other hair dye compounds. In a gradient of LC conditions, either 24 [13] or 40 [14] hair dye compounds, including 4A3NP, were simultaneously analyzed. To develop and validate the LC-MS method for 4A3NP analysis, these analysis methods were modified and consulted.

## 2. Materials and Methods

### 2.1. Chemicals

Both 4-amino-3-nitrophenol (4A3NP) and 2-aminophenol (2AP) were purchased from Sigma Aldrich Co. (St. Louis, MO, USA). Acetonitrile, methanol, and HPLC-grade water were obtained from Honeywell Burdick & Jackson Co. (St. Harvey, MI, USA). Formic acid was obtained from Merck Milipore Co. (Kenilworth, NJ, USA).

### 2.2. LC-MS/MS Analysis Conditions

The LC-MS/MS analysis of 4A3NP was conducted using an Agilent 1290 system (Agilent Technology, Waldbronn, Germany) coupled with an API 4000 mass spectrometer (AB Sciex, Massachusetts, USA). The chromatographic column employed was a LUNA 3 μm C18 150 × 3.00 mm column (Phenomenex, CA, USA) with a guard column (SecurityGuard Cartridges RP-1, 4 × 3.0 mm, Phenomenex, CA, USA) maintained at a temperature of 40 °C. An isocratic system consisting of 40% acetonitrile in water with 0.1% formic acid (*v*/*v*) was utilized for a duration of 5 min. The injection volume was set to 10 μL, and the flow rate was maintained at 0.3 mL/min.

For multiple reaction monitoring (MRM), the positive mode was employed with the following transitions set: *m*/*z* 154.9 → 137 for 4A3NP and *m*/*z* 109.9 → 92 for 2AP (internal standard (IS)) (Table 1). The mass spectrometry system was configured with the following parameters: the IonSpray voltage (IS) set to 5000 V, the ion source temperature maintained at 350 °C, ion source gases 1 (GS1) and 2 (GS2) flowing at 50 and 55 L/h, respectively, the curtain gas (CUR) set to 20 L/h, the entrance potential (EP) set to 10 V, and the collision gas (CAD) set to 4 L/h.

### 2.3. Method Validation

The solution of extraction and dilution was methanol, except for IS solution (2AP in ethanol), which showed a more symmetrical and higher peak shape than that in methanol (Appendix A). The stock solutions of 4A3NP were prepared to the amount of 10 mg/mL in methanol. To analyze the linearity of 4A3NP, each calibration standard solution (15 μL) in methanol was mixed with 135 μL of the blank matrix of swab wash (WASH), stratum corneum (SC), SKIN (dermis + epidermis), and receptor fluid (RF). The final working solutions were 20, 50, 80, 100, 200, 500, 800, and 1000 ng/mL. Then, 50 μL of calibration solution was mixed with 150 μL of IS solution in ethanol (100 ng/mL). The mixtures were vortexed and centrifuged at 21,000× *g* for 10 min. Finally, the supernatant was filtered using a 0.2 μm polytetrafluoroethylene (PTFE) filter (ADVANTEC, Dublin, California, USA), and 10 μL of the supernatant was injected for LC-MS/MS. All samples were stored at −20 °C before LC-MS/MS analysis.

To determine the intra- and interday accuracy and precision of 4A3NP, five replicates were examined at four-point quality control (QC) sample levels (LLOQ, low, middle, and high QC). Because of the matrix effect (Figure 1, double blank), the LLOQ of WASH and SC was set to 80 ng/mL, which was a higher level than those of SKIN and RF (50 ng/mL). Thus, for the SKIN and RF matrix, the concentration of the QCs was 50, 150, 400, and 750 ng/mL, and 80, 240, 400, and 750 ng/mL were set for the QCs of WASH and SC. The intraday precision of 4A3NP was calculated by the relative standard deviation (RSD) of the QC samples, and the interday precision was determined by the analysis of the QC samples on three consecutive days [15]. The accuracy and precision were calculated based on the following formula:% Accuracy = (measured concentration/nominal concentration) × 100
% Precision = (measured standard deviation/nominal concentration) × 100

### 2.4. In Vitro Dermal Absorption

In the skin absorption study of 4A3NP, a static Franz diffusion cell system (Hanson, Chatsworth, California, USA) was utilized. The mini-pig skins, measuring 500 μm in thickness (1.5 × 1.5 cm), were obtained from Apures (Pyeongteak, Repubic of Korea) and stored at −20 °C before use. In accordance with the guidelines of the Ministry of Food and Drug Safety [8], the receptor fluid (RF) solution comprised 6% polyethylene glycol 20 oleyl ether (PEG), which was chosen considering the solubility (Log Pow) of 4A3NP (Table 1), being suitable for hydrophobic chemicals. The mini-pig skins, aged 6 months (*n* = 4 for oxidative and *n* = 6 for non-oxidative conditions), were placed on each cell and maintained at a temperature of 32 ± 1 °C. For the oxidative condition of hair dye, 4A3NP (3%) and hydrogen peroxide (6%) were mixed at a 1:1 ratio in following the Korean dyeing standard [16], resulting in a concentration of 1.5% 4A3NP upon application. For the non-oxidative conditions, it was diluted to a final concentration of 1% using distilled water before application. Subsequently, 17.7 µL of each sample was applied to the mini-pig skin (10 μL/cm^2^). As per general hair dye conditions, the skin was wiped with an alcohol swab after 30 min of application (WASH_30 min) and replaced on each cell. After 24 h of application, the skin was wiped again with an alcohol swab (WASH_24 h). To remove the stratum corneum of the skin, the tape (Scotch™, 3M, Maplewood, MN, USA) was cut into 1.5 × 1.5 cm pieces and stripped 15 times (SC). Then, the mini-pig skin was cut into 8 pieces (SKIN). All samples except those for the WASH matrix were placed into 10 mL of 50% methanol in water (*v*/*v*), with the WASH matrix requiring 20 mL of solvent, and sonicated for 1 h. RF samples were collected at 0, 1, 2, 4, 8, 12, and 24 h and stored in a refrigerator before analysis. For the analysis of 4A3NP, 50 μL of each extracted sample was mixed with 150 μL of ethanol (containing IS) and centrifuged at 21,000× *g* for 10 min. The supernatant was filtered using a 0.2 μm polytetrafluoroethylene (PTFE) filter (ADVANTEC, Dublin, CA, USA) and stored at −80 °C before LC-MS/MS analysis. The samples that exceeded the calibration range (50–1000 ng/mL) were diluted with the respective blank matrix and then reanalyzed.

### 2.5. Statistical Analysis

All data are presented as the mean ± standard deviation (SD). Statistical calculations were performed using Excel 2013 (Microsoft for Windows) and GraphPad Prism software version 5.04 (San Diego, CA, USA).

## 3. Results

### 3.1. Validation of Analytical Methodology (Accuracy and Precision)

The LLOQ levels (50 ng/mL for WASH and SC and 80 ng/mL for SKIN and RF) of 4A3NP are presented in Figure 1. The peak heights of 4A3NP in the matrix samples were three-fold higher than the signal-to-noise ratio (S/N), and the shapes were symmetrical. The retention times of 4A3NP and 2AP were 3.2 and 1.65 min, respectively. The calibration curve of 4A3NP was strongly linear within the range of examination (20–1000 ng/mL) (R^2^ = 0.9962–0.9993) (Table 2 and Appendix A).

Five replicates of 4A3NP QC samples determined the intra- and interday accuracies and precisions at four concentrations (LLOQ, low, middle, and high QC of 50 (or 80), 150 (or 240), 400, and 750 ng/mL, respectively) in each matrix. The intraday accuracies of WASH, SC, SKIN, and RF were 96.29~103.3%, 93.5~99.2%, 102.4~111.73%, and 100~104.64%, and their precisions were 1.7~7.51%, 3.42~6.94%, 2.41~7.47%, and 2.4~14.46% based on the coefficient of variation (CV), respectively. The interday accuracies of WASH, SC, SKIN, and RF were 96.29~103.3, 97.29~104.02%, 100.17~102.62%, and 100.35~102.37%, and their precisions were 1.7~7.51%, 5.86~7.63%, 2.85~8.42%, and 1.84~10.32% based on the CV, respectively (Table 2). Evaluation of the carryover, which pertains to the unintended transfer of a substance from one analytical sample to the next, subsequent to the administration of a high-concentration sample (750 ng/mL), did not impact any of the matrices (Appendix A). These developed methods showed well-fitted accuracy and precision values under the validation guideline (within ± 20% of the LLOQ and ±15% of the nominal concentration) [15]. Thus, the current analytic method was used for the quantitation of 4A3NP.

### 3.2. In Vitro Dermal Absorption

The amounts of 4A3NP in WASH_30 min, WASH_24 h, SC, SKIN, and RF after the dermal absorption study are shown in Figure 2. As a result of quantitation of 4A3NP in various matrices, WASH_30 min was mainly quantitated in the ranges of 88.51 ± 3.20% and 93.39 ± 6.31% in oxidative and non-oxidative hair dye conditions, respectively (Appendix A). Meanwhile, the smallest amount of 4A3NP was detected in the stratum corneum (SC) as follows: 0.14% ± 0.23% (oxidative) and 0.05% ± 0.08% (non-oxidative) (Appendix A). In addition, the total recoveries in the oxidative and non-oxidative samples were 92.989% ± 3.42% and 100.30% ± 4.08%, respectively. Under the different concentrations in the oxidative (1.5%) and non-oxidative (1.0%) conditions, the total absorbed 4A3NP in accordance with the application dose for SKIN and RF was 2.83% ± 1.48% and 5.62% ± 2.19%, respectively. Permeation values were calculated from the cumulative 4A3NP in the receptor fluid at each sampling point and normalized to the exposed skin surface area (1.77 cm^2^) (Figure 3). In the finite dose application, the concentrations of 4A3NP were gradually increased in both conditions until 12 h, when there were rarely any changes until 24 h. The calculated equilibrium fluxes (Js) in the oxidative and non-oxidative conditions were 0.16 ± 0.07 µg/cm^2^/h and 0.23  ±  0.09 µg/cm^2^/h, respectively (Table 3).

## 4. Discussion

To analyze and quantify 4A3NP, the LC-MS method of Fu et al. was modified [11]. MRM is an analytical method that tracks a daughter ion (fragment of a parent ion) from a parent ion during the analysis time. It is a particular and sensitive mass spectrometry technique for analyzing a target metabolite within a complex mixture [17,18]. Given that matrices such as WASH, SKIN, SC, and RF in the in vitro dermal absorption study contain numerous other substances that may cause interference or disruption in detecting the target compound, this is an advantageous way to use MRM, which recognizes only the specific transition pattern of the target compound. Therefore, before establishing the analytical method for 4A3NP, transitions of 4A3NP and 2AP were analyzed using the MRM technique. Due to the neutral loss of H_2_O, protonated 4A3NP in quadrupole 1 (Q1) was fragmented to *m*/*z* 137 in quadrupole 3 (Q3). Additionally, 2AP, an internal standard, was also fragmented to *m*/*z* 92, water loss product, in Q3.

In the validation results, the linearity (R^2^) of each matrix exceeded 0.99, and the accuracy and precision of the quality control (QC) samples in each matrix met the guidelines of the Ministry of Food and Drug Safety (MFDS) (within ±20% of the lower limit of quantification (LLOQ) and ±15% of the nominal concentration) [15]. Thus, this analytical method demonstrated its robustness and reliability for quantifying 4A3NP in each matrix sample.

In vitro skin absorption of 4A3NP was previously conducted using human dermatomed skin by Toner [4]. In Toner’s study, a homogeneous cream formulation of ^14^C-labeled 4A3NP was applied to human breast skin (400 μm) using a flow-through diffusion system for 24 h. Under oxidative (1.5%) and semi-permanent conditions (1%), 20 mg/cm^2^ of the sample was loaded, and after 30 min and 24 h, any remaining formulation on the skin was rinsed off [4]. In contrast to Toner’s experiment, in this study, 10 mg/cm^2^ of 4A3NP was applied (which is two-fold lower than Toner’s amount). Im et al. (2021) reported that the application dose affects the percentage of dermal absorption, indicating a higher level of dermal absorption with a lower application dose, and suggested using 10 mg (or µg)/mL as the application dose [18]. Although the concentration of 4A3NP was the same as in Toner’s experiment, the absorption rate (%) was approximately 2-fold and 14-fold higher for the oxidative and non-oxidative conditions, respectively, because the application volume in this study was lower. Moreover, compounds generally exhibit slightly increased permeability in pig skin compared with human skin [19]. Hence, it appears that this dermal absorption study demonstrated a higher absorption rate. In the risk assessment document of 4A3NP [2], the SCCP pointed out the high variability of skin penetration in Toner’s study. Barbero and Frasch (2009) also presented a high intraspecies coefficient of variation in humans compared with pigs [20], and Qvist et al. (2000) reported that this may be attributed to the selected skin site, age, and lifestyle of the donor [21]. Consequently, many papers have suggested pig skin as a surrogate for human skin due to its physiological and biochemical similarity, as well as its high reproducibility [19,20]. Therefore, from the perspective of cosmetic risk assessment, it may be more appropriate to apply 10 mg/cm^2^ and use pig skin for the dermal absorption of 4A3NP to consider the worst-case scenario.

In the results of dermal absorption (%) of 4A3NP, the amount absorbed under oxidative hair dye conditions was lower than that under non-oxidative conditions. Even when considering the application amount per unit area, with 100 ug/cm^2^ for the non-oxidative conditions (1%) and 150 ug/cm^2^ for the oxidative (1.5%) conditions as the active ingredients, this study also demonstrated a slightly lower absorption rate in the oxidative case. The permeation profile of 4A3NP in non-oxidative conditions was generally higher (Figure 3), and the permeation parameter (Js) between both conditions also showed similar results (Table 3). Hydrogen peroxide induces oxidative damage in the skin, and high concentrations of it can trigger apoptosis and cell necrosis in skin cells [22]. Furthermore, reactive oxygen species (ROS) originating from hydrogen peroxide can inflict harm by interacting with nucleic acids, proteins, and lipids, leading to functional loss and tissue damage [22]. In a dermal absorption study of hair dye material, the absorption rate of the formulation containing the developer’s mix including hydrogen peroxide (oxidative conditions) was lower than that of the formulation without the developer’s mix [23]. The exact mechanism is unknown, but it is thought that exposure to hydrogen peroxide may interfere with skin penetration.

A structural isomer of 4A3NP, 2-amino-5-nitrophenol (2A5NP), was conducted with dermal absorption in the oxidative hair dye conditions (1.5%) [24] and demonstrated higher dermal absorption (13.6 ± 2.9%) compared with this study on 4A3NP (2.83 ± 1.48%). Skin absorption occurs through passive diffusion and is significantly influenced by the formulation and physical properties of the test chemicals [25]. Among these properties, the molecular weight (MW) and logarithmic octanol-water partition coefficient (Log Pow) are key factors for predicting skin absorption (permeability) [26]. Generally, compounds with a low molecular weight (<150 Da) and Log Pow levels between 1 and 2 exhibit high absorption rates [27]. The estimated Log Pow of 2A5NP was 1.5, while 4A3NP was more soluble in water (Log Pow: 0.41). Therefore, this study presented a lower absorption rate (%) of 4A3NP under oxidative conditions. In conclusion, the analytical method for quantitating 4A3NP in various matrices was developed, and the absorption rates of 4A3NP were determined to be 2.83 ± 1.48% (4.24 ± 2.21 μg/cm^2^) and 5.62 ± 2.19% (5.62 ± 2.19 μg/cm^2^) in oxidative and non-oxidative conditions, respectively.

## 5. Conclusions

An analytical method for examining 4A3NP using LC-MS/MS was validated and quantitated in various matrices. These developed methods showed well-fitted linearity (R^2^ = 0.9962–0.9993), accuracy (93.5–111.73%), and precision (1.7–14.46%) when following the validation guidelines. The total dermal absorption rates of 4A3NP in oxidative (1.5%) and non-oxidative (1%) conditions were determined to be 2.83 ± 1.48% (4.24 ± 2.21 μg/cm^2^) and 5.62 ± 2.19% (5.62 ± 2.19 μg/cm^2^), respectively.

## Figures and Tables

**Figure 1 toxics-12-00340-f001:**
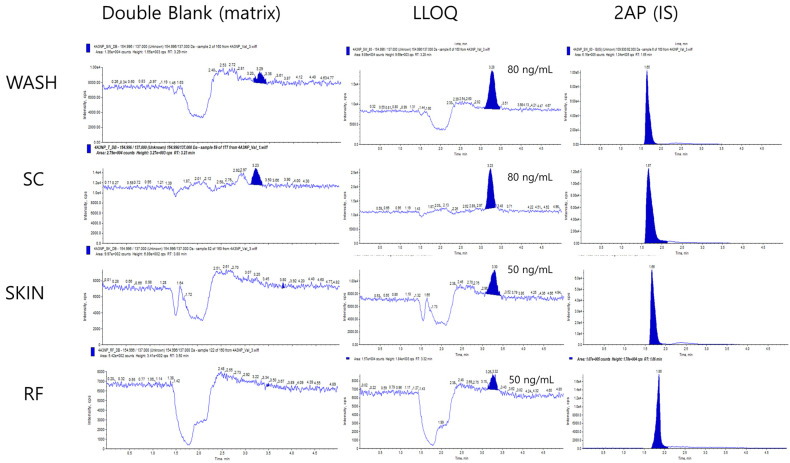
Chromatograms of 2-aminophenol (IS) and 4-amino-3-nitrophenol at the double blank and LLOQs in alcohol swab (WASH), stratum corneum (SC), dermis and epidermis (SKIN), and receptor fluid (RF) (6% polyethylene glycol).

**Figure 2 toxics-12-00340-f002:**
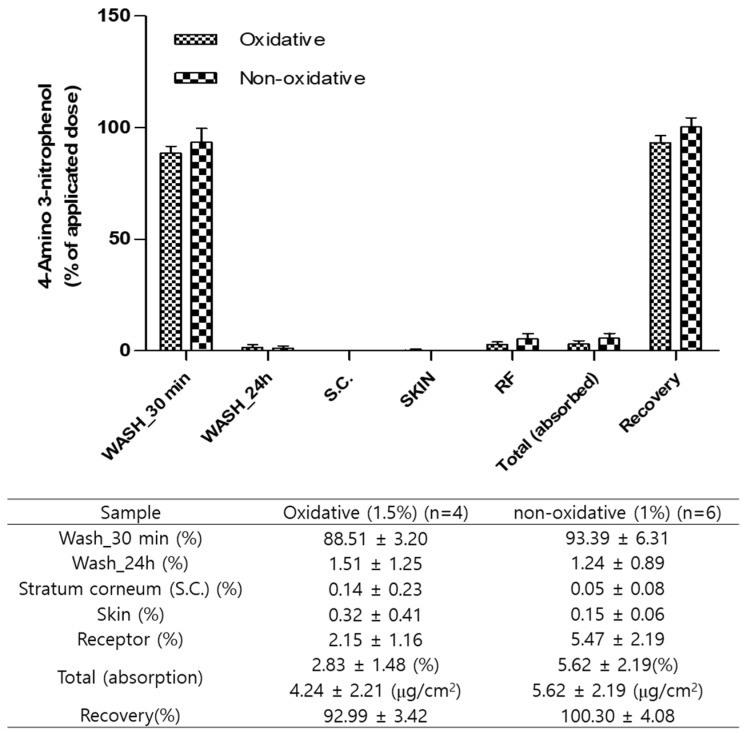
In vitro dermal absorption rate of 4-amino-3-nitrophenol. WASH = alcohol swab; SC = stratum corneum; SKIN = dermis and epidermis; RF = receptor fluid (6% polyethylene glycol).

**Figure 3 toxics-12-00340-f003:**
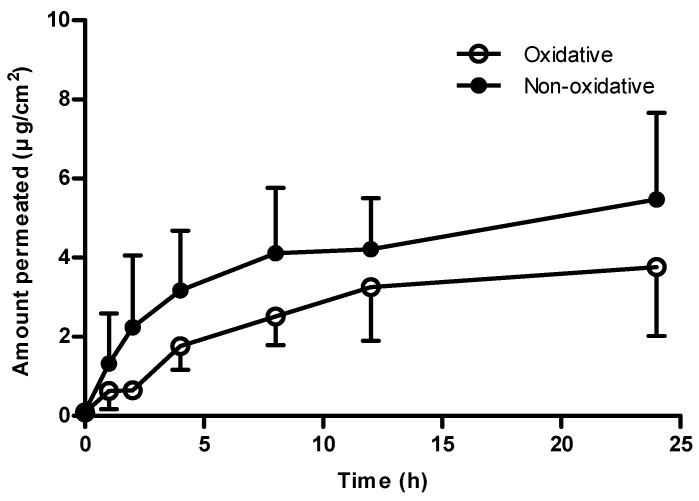
Percutaneous penetration profile of 4A3NP under oxidative (○) and non-oxidative (●) conditions. Values are presented as penetration amount through excised pig skin area (cm^2^).

**Table 1 toxics-12-00340-t001:** Physicochemical properties and mass transitions of 4-amino-3-nitrophenol (4A3NP) and 2-aminophenol (2AP).

Name	MW	CAS No.	Chemical Composition	Log Pow	Structure	MS Transition (Positive)
Q1	Q3	CE
4-amino-3-nitrophenol(4A3NP)	154.12	610-81-1	C_6_H_6_N_2_O_3_	0.41	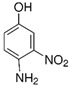	154.9	137	15
2-aminophenol(2AP)	109.13	95-55-6	C_6_H_7_NO	0.62	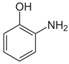	109.9	92	23

**Table 2 toxics-12-00340-t002:** Intra- and interday accuracy and precision of 4-amino-3-nitrophenol in each matrix (*n* = 5).

Compound	Conc. (ng/mL)		Intraday (%)			Interday (%)	
		Mean ± SD	Accuracy	Precision	Mean ± SD	Accuracy (%)	Precision (%)
WASH (R^2^ = 0.9987)	80	79.74 ± 1.77	99.68	2.23	79.89 ± 4.92	99.68	2.23
	240	237.40 ± 4.04	98.92	1.70	238.60 ± 16.80	98.92	1.70
	400	413.20 ± 16.22	103.30	3.93	411.33 ± 36.16	103.30	3.93
	750	722.20 ± 54.27	96.29	7.51	734.2 ± 77.25	96.29	7.51
SC (R^2^ = 0.9993)	80	74.80 ± 5.19	93.50	6.94	77.83 ± 4.57	97.29	5.88
	240	224.40 ± 7.67	93.50	3.42	240.86 ± 18.38	100.36	7.63
	400	389.80 ± 26.36	97.45	6.76	405.53 ± 23.78	101.38	5.86
	750	744.00 ± 31.01	99.20	4.17	780.13 ± 48.17	104.02	6.17
SKIN (R^2^ = 0.9962)	50	51.20 ± 2.18	102.40	4.25	50.08 ± 1.42	100.17	2.85
	150	154.80 ± 11.56	103.20	7.47	151.06 ± 9.21	100.71	6.10
	400	445.40 ± 10.78	111.35	2.42	410.46 ± 27.57	102.62	6.72
	750	838.00 ± 20.17	111.73	2.41	752.80 ± 63.38	100.37	8.42
RF (R^2^ = 0.9991)	50	52.32 ± 7.56	104.64	14.46	50.19 ± 5.18	100.39	10.32
	150	150.00 ± 5.10	100.00	3.40	153.26 ± 7.91	102.18	5.16
	400	407.60 ± 9.79	101.90	2.40	409.46 ± 7.54	102.37	1.84
	750	761.60 ± 24.93	101.55	3.27	752.60 ± 22.41	100.35	2.98

WASH = alcohol swab; SC = stratum corneum; SKIN = dermis and epidermis; RF = receptor fluid (6% polyethylene glycol).

**Table 3 toxics-12-00340-t003:** Permeation parameter of 4-amino-3-nitrophenol through excised mini-pig skin (mean ± standard deviation).

Substance	Conditions	Permeation Parameter Js (Equilibrium Flux, μg/cm^2^/h)
4A3NP	Oxidative (1%)	0.16 ± 0.07
	Non-oxidative (1.5%)	0.23 ± 0.09

## Data Availability

The original contributions presented in the study are included in the article/Appendix A, further inquiries can be directed to the corresponding author.

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
