# Peer review of "Analytical Method Development and Dermal Absorption of 4-Amino-3-Nitrophenol (4A3NP), a Hair Dye Ingredient under the Oxidative or Non-Oxidative Condition"

_toxics, 2024, doi:10.3390/toxics12050340_

Round 1

Reviewer 1 Report

Comments and Suggestions for Authors

The manuscript entitled” Analytical method development and dermal absorption of 4-amino-2 3-nitrophenol (4A3NP), a hair dye ingredient under the oxidative or 3 non-oxidative condition” reports method validation and in vitro dermal absorption study of 4A3NP. The results are clear and concisely summarized. The methods used in this study is proper and well established in the related field. Moreover, the data were submitted to statistical analysis.

Author Response

Thank you for review this paper. 

Reviewer 2 Report

Comments and Suggestions for Authors

The article titled " Analytical method development and dermal absorption of 4-amino- 3-nitrophenol (4A3NP), a hair dye ingredient under the oxidative or non-oxidative condition " aimed to validate an analytical method and conduct an in vitro dermal absorption study of 4A3NP. However, the aim was not clearly articulated in the introduction. The thesis presented in the article is intriguing and warrants further investigation. The Authors made commendable efforts to perform validation experiments on various matrices in accordance with international validation guidelines. Nevertheless, several issues require attention. The most significant concern is the lack of any conclusive inference drawn from the experiments. The "Conclusion" section merely summarizes the results without offering any insights into whether the observed permeability poses a risk to humans.

1.       All results of the method validation, such as matrix factor, dilution integrity (if applicable), carry-over, calibration curve parameters for each matrix, and analyte stability, should be thoroughly presented. Additionally, it would be beneficial to know if any weighting of the calibration curve points was performed.

2.       The abstract contains excessive methodological details and lacks a succinct conclusion.

3.       Some sentences are linguistically challenging and may benefit from revision by a native speaker for clarity and grammatical accuracy.

4.       It would be helpful to include an explanatory comment at the end of the paragraph on line 51 to elucidate the rationale behind the chosen research approach.

5.       Clarification is needed regarding the specific differences referred to on line 79.

6.       The method of sample preparation in section 2.4 requires clarification, particularly regarding the volume of methanol-water used.

7.       In Table 2, rounding of results should be adjusted to reflect uncertainties accurately.

8.       Chromatograms of real samples (at the lowest and highest concentrations) should be added to the Supplementary Information.

9.       Information regarding whether the obtained results fall within the range of the standard curve and the minimum and maximum detected concentrations [ng/ml] in each matrix is essential for method applicability.

10.  Table 2 requires appropriate rounding of values. The result and uncertainty for S.C. are provided with insufficient significant figures, while "wash 30 min" is rounded insufficiently.

11.  The first paragraph on page 8 should elaborate on the errors in skin pretreatment processes, justify the suitability of a 10 mg/ml concentration over 5 mg/ml, and clarify why pig skin is deemed appropriate despite the lack of correlation with human skin.

12.  Line 263-265 requires clarification regarding whether the same model was utilized for the mentioned study.

Author Response

1. All results of the method validation, such as matrix factor, dilution integrity (if applicable), carry-over, calibration curve parameters for each matrix, and analyte stability, should be thoroughly presented. Additionally, it would be beneficial to know if any weighting of the calibration curve points was performed.

->  Thank you for the comment. I added supplement data for carry-over and calibration curve parameters (weight of the calibration curve in Fig. S1).  

2. The abstract contains excessive methodological details and lacks a succinct conclusion.

-> Thank you for the comment. I revised it. 

3. Some sentences are linguistically challenging and may benefit from revision by a native speaker for clarity and grammatical accuracy.

-> Thank you for the comment. I revised it. 

4. It would be helpful to include an explanatory comment at the end of the paragraph on line 51 to elucidate the rationale behind the chosen research approach.

-> Thank you for the comment. I revised it. 

5. Clarification is needed regarding the specific differences referred to on line 79.

-> Thank you for the comment. I revised it. 

6. The method of sample preparation in section 2.4 requires clarification, particularly regarding the volume of methanol-water used.

-> Thank you for the comment. I revised it. 

7. In Table 2, rounding of results should be adjusted to reflect uncertainties accurately.

-> Thank you for the comment. I revised it. 

8. Chromatograms of real samples (at the lowest and highest concentrations) should be added to the Supplementary Information.

-> Thank you for the comment. I added it to the supplement data file. 

9. Information regarding whether the obtained results fall within the range of the standard curve and the minimum and maximum detected concentrations [ng/ml] in each matrix is essential for method applicability.

-> Thank you for the comment. I added it to the supplement data file. 

10. Table 2 requires appropriate rounding of values. The result and uncertainty for S.C. are provided with insufficient significant figures, while "wash 30 min" is rounded insufficiently.

-> Thank you for the comment. I revised it. 

11. The first paragraph on page 8 should elaborate on the errors in skin pretreatment processes, justify the suitability of a 10 mg/ml concentration over 5 mg/ml, and clarify why pig skin is deemed appropriate despite the lack of correlation with human skin.

-> Thank you for the comment. I revised it. 

12. Line 263-265 requires clarification regarding whether the same model was utilized for the mentioned study.

-> Thank you for the comment. I revised it. 

Reviewer 3 Report

Comments and Suggestions for Authors

The paper introduced the detection method and dermal absorption of 4A3NP. The paper would provide some data basis for conducting relevant research, but the paper is lack of innovation.

(1) As a hair dye, 4A3NP is not a new substance, so there has been many research about its detection methods and Risk assessment. The introduction section of this paper has not introduced any innovative changes that have been made compared to previous studies.

(2) Method development should have a conditional optimization process, but this paper has not described the process.

(3) There are too few new references, so more references from the past 5 years should be added

Comments on the Quality of English Language

The paper introduced the detection method and dermal absorption of 4A3NP. The paper would provide some data basis for conducting relevant research, but the paper is lack of innovation.

(1) As a hair dye, 4A3NP is not a new substance, so there has been many research about its detection methods and Risk assessment. The introduction section of this paper has not introduced any innovative changes that have been made compared to previous studies.

(2) Method development should have a conditional optimization process, but this paper has not described the process.

(3) There are too few new references, so more references from the past 5 years should be added

Author Response

(1) As a hair dye, 4A3NP is not a new substance, so there has been many research about its detection methods and Risk assessment. The introduction section of this paper has not introduced any innovative changes that have been made compared to previous studies.

-> Thank you for the comment. I revised it. 

(2) Method development should have a conditional optimization process, but this paper has not described the process.

-> Thank you for the comment. I revised it.  

(3) There are too few new references, so more references from the past 5 years should be added

-> Thank you for the comment. I added new references. 

Round 2

Reviewer 3 Report

Comments and Suggestions for Authors

no comments

Comments on the Quality of English Language

no comments